# Moderating effects of resilience and self-esteem on associations between self-reported oral health problems, quality of oral health, and mental health among adolescents and adults in Nigeria

**Olanrewaju Ibikunle Ibigbami**[1]*, **Morenike Oluwatoyin Folayan**[2,3,4,5], **Olakunle Oginni**[1], **Joanne Lusher**[6], **Nadia A. Sam-Agudu**[7,8]

1 Department of Mental Health, Obafemi Awolowo University, Ile-Ife, Nigeria, 2 Department of Child Dental Health, Obafemi Awolowo University, Ile-Ife, Nigeria, 3 Nigeria Institute of Medical Research, Yaba, Lagos State, Nigeria, 4 Community Oral Health Department, Tehran University of Medical Sciences, Tehran, Iran, 5 Faculty of Health Sciences, University of Zaragoza, Zaragoza, Spain, 6 Provost's Group, Regent's University, London, United Kingdom, 7 International Research Center of Excellence, Institute of Human Virology Nigeria, Abuja, Nigeria, 8 Institute of Human Virology, University of Maryland School of Medicine, Baltimore, MD, United States of America

* oibigbami@oauife.edu.ng

## Abstract

### Background

There is an intersection between oral and mental health though the studies on these intersections are few. This study investigated associations between self-reported oral health problems, quality of oral health, and depression and general anxiety among adolescents and adults in Nigeria; and analysed the moderating effects of resilience and self-esteem on these associations.

### Methods

In this secondary analysis, data were extracted from the database of an online survey conducted among participants 13 years and older and living in Nigeria about their self-reported psychological wellbeing. The data was collected between September and October 2020. Dependent variables were self-reported presence of oral health problems (yes/no) and self-reported quality of oral health (using a five-item scale ranging from "very good" to "very poor"). Independent variables were depressive and anxiety symptoms. Moderating factors evaluated were resilience and self-esteem. Multivariable logistic regression analysis was used to determine the associations between the dependent and independent variables after adjusting for confounders (age, sex, employment status and educational status). A path analysis was conducted to determine the moderating effects of self-esteem and resilience on associations between dependent and independent variables.

**Data Availability Statement:** All relevant data are within the paper and its Supporting Information files.

**Funding:** The authors received no specific funding for this work.

**Competing interests:** The authors have declared that no competing interests exist.

## Results

We extracted data for 2,757 adolescents and adults aged 13 to 62 years, of which 2,062 (74.8%) reported having oral health problems and 925 (33.6%) reported poor quality of oral health. Higher levels of depressive symptoms were significantly associated with higher odds of oral health problems (AOR: 1.07; 95% CI: 1.04–1.10; p<0.001). Higher levels of depressive symptoms (AOR: 1.05; 95% CI: 1.03–1.07; p<0.001), and higher levels of anxiety symptoms (AOR: 1.07; 95% CI: 1.04–1.11; p<0.001) were significantly associated with poor quality of oral health. Resilience significantly moderated the association between anxiety symptoms and oral health problems (AOR = -0.004; 95% CI: -0.006 –-0.001; p = 0.002).

## Conclusion

Depression may be a risk indicator for self-reported oral health problems, while depression and anxiety appear to be risk indicators for self-reported poor quality of oral health. These factors could be included as confounders in future studies on oral health problems and quality of oral health among adolescents and adults in Nigeria.

## Introduction

Oral health affects the quality of life [1], and optimal oral health contributes to individual well-being [2]. Globally, the growing burden of oral health problems poses a major problem for healthcare managers [3]. This is exacerbated by a shortage of dental services and poor utilization of these services even where available [4] mainly due to high costs [5, 6]. The poor oral health resulting from poor use of preventive dental services, is associated with mental health distress across all income levels, globally [7–10].

The relationship between mental health and oral health is bi-directional. Depression is a risk factor for poor oral health [11], because it leads to the neglect of oral care, and poor oral health is a depression medication-related adverse event [12, 13]. Furthermore, poor oral health can worsen psychological distress and depression because of the poor perception of one's oral health [14] and diminished self-worth [15]. Anxiety is another mental health disorder that can also increase the risk for poor oral health [16, 17], though this association is not as consistent as that observed with depression [18]. There is a gap in the evidence base on the bi-directional relationship between mental and oral health in many African countries. For example, in Nigeria, there are multiple studies on the impact of oral diseases on the oral health-related quality of life [19–22] among different populations [23, 24], However, few studies have generated evidence on associations between mental health and oral health in populations in Africa [25–27], and none had assessed the impact of mental health on self-perception about the quality of oral health.

Self-perception of oral health problems may increase the utilization of dental care services. Dental healthcare service utilization in Nigeria is, however, low [28, 29] and mainly accessed for curative purposes [30]. This may be because most adults in Nigeria perceive their oral health status to be good [31]. Furthermore, psychological health affects personal views and perspectives about health and quality of life [32, 33]. There is, however, no available data on factors that affect perceptions about one's oral health status and quality of oral health.

The prevalence of psychological distress among adults in Nigeria is as high as 69.9% [34]. Also, between 28.5% and 68.4% of adults have untreated oral health conditions [35].

Nonetheless, there is a lack of evidence on the mechanisms by which mental health may affect individual perceptions of oral health problems, the negative impact of these problems on one's quality of life and if these interactions can increase the uptake of dental care services for the treatment of oral health conditions. This is despite the evidence of the strong predictive power of self-reported health for prospective health and social outcomes [36].

In addition, resilience and self-esteem can protect against adverse mental health outcomes following adverse exposures, and moderates the association between poor oral health and mental health difficulties [37]. Resilience describes an individual's innate ability to positively adapt in the face of stress and adversity [37]. On the other hand, self-esteem, which is the evaluative and affective dimension of the self-concept (a cognitive schema that sums one's beliefs and knowledge about their personal attributes and qualities) [38], affects reactions to stressful events and how individuals cope with stress [39]. Thus, resilience can minimize the impact of health-related adversities [40, 41], including those arising from poor oral health [42, 43], thereby attenuating the associations between poor oral health and mental health disorders like depression and anxiety [44]. There is however, little known about how individual resilience and self-esteem moderate oral health in Nigeria.

This study, therefore, sought to determine associations between self-reported oral health problems, self-reported quality of oral health, and mental health disorders (depressive and generalized anxiety symptoms) among adolescents and adults in Nigeria. We also investigated whether resilience and self-esteem acted as moderating variables in these associations. We hypothesize that the presence of oral health problems and perceived poor quality of oral health will positively predict the occurrence of depression and anxiety, while resilience and self-esteem will enhance the associations between self-reported oral health problems, poor quality of oral health and depression and anxiety among the study population.

## Methods

### Ethical considerations

Ethical approval was obtained from the Health Research Ethics Committee of the Institute of Public Health at Obafemi Awolowo University in Ile-Ife, Nigeria (IPHOAU/12/1571). Our online survey recruited adolescents and adults 13 years and above who were resident in Nigeria. All participants provided consent for study participation by ticking a checkbox to indicate consent after reading the informed consent sheet. Participants who did not consent to participate were taken to the end of the survey and thanked for their time. Parental consent was waived for minors under the legal age of 18 years, since this was a non-invasive online study. The waiver was based on the national health research guidelines on sexual and reproductive health involving minors that notes: 'To participate in non-therapeutic research, persons aged 9 years and under require only parental consent while persons aged 10–12 years require parental consent as well as assent from the young people. However, persons aged 13 and above and emancipated minors can consent for themselves without parental consent' [45]. All study participants who took the online survey were offered compensation of 100 Naira (~$0.27) worth of internet data.

**Community engagement in study implementation.** The primary study was conceptualised by Total Health Empowerment and Development Initiative (THEDI), a non-governmental organisation that provides healthcare services to populations highly vulnerable to HIV in Nigeria. The aim of the primary study was to identify the magnitude of mental health challenges faced by the community served by THEDI when compared to the general population and thereby, develop an advocacy agenda to address these concerns.

**Study design, setting, and population.** This was a secondary analysis of data collected from a nation-wide survey. The target population were people age 13 years old and older. Data on the sexual, oral, mental, and general health of study participants were collected via an online electronic survey (SurveyMonkey). Survey Monkey complies with both the European Union-United States and Swiss-United States Privacy Shield programs [46]. The survey was opened to the public between 16th September and 31st October 2020.

**Sample size.** The sample size for the primary study was estimated at 1100 participants using the formula proposed by Kline [47]. The study questionnaire included 11 variables with an estimated maximum of 55 parameters in multivariate analyses. Based on the high rates of incomplete responses with online surveys (up to 50% from previous experience) and the need for subgroup analysis, the minimum sample size for the primary study was increased to 3000 (500 per proposed study group and 1000 for a comparison group). In total, the primary study recruited 3,529 participants drawn from the 36 States and the Federal Capital Territory in Nigeria.

**Recruitment of study participants.** A 3-day online training was conducted for 23 THEDI peer educators and six organisational study staff. The team was trained on questionnaire administration, communication with participants, ethics of research engagement and the study implementation process. The team recruited study participants through two non-probabilistic, purposive sampling methods–venue-based and/or snowball. For venue-based sampling, an open link was provided in the THEDI office and offices of its partners, to enable target populations with low literacy fill the study questionnaire with support from onsite, in-person peer educators. For snowball sampling, respondents who completed the survey were encouraged to recruit peers as participants. Additional participants were recruited through an online river sampling method by sharing the survey link with contacts on social media (including Facebook, Twitter, WhatsApp groups and Instagram) and network email lists.

**Study instrument.** The purpose of the study was explained in the introductory section of the survey questionnaire. The introductory section also informed participants that their participation was voluntary and that all information obtained would be managed confidentially. Each participant was able to freely edit their responses until they chose to submit. The survey was administered in English and contained instruments validated for collecting mental health and oral health survey data. The instruments were the Patient Health Questionnaire (PHQ-9), the Generalised Anxiety Disorder (GAD-7) scale, the Connor-Davidson resilience scale and the Rosenberg self-esteem scale.

The questionnaire was reviewed and revised for the validity of its content. The revised questionnaire was then reviewed by the six THEDI non-government organisation staff and after that, by the 23 peer educators. The final version was pretested with 40 individuals. The online questionnaire was designed to be completed anonymously, and did not install any tracker cookies on respondents' devices. IP addresses were decoupled and encrypted automatically while the survey was active online, making it impossible to provide further direct information to participants after completion of the questionnaire.

## Data variables

**Dependent variables.** *Self-reported oral health problems*. This was a measure of the perception individuals had about having oral health diseases [48] as assessed using a questionnaire that evaluated 10 oral health problems, namely: hole in the tooth, sensitive tooth, bleeding gums, swollen gums, bad breath, fractured tooth, discoloured tooth, painful tooth, oral ulcers, and missing tooth. Respondents checked a box if they had any of the oral health problems. They also had the option of writing in other oral health problems not listed. These 10 checklist

questions enabled participants to answer the next question more objectively. The next question inquired if respondents felt they had any oral health problems (yes or no). Self-reported oral health problems have been accepted as a valid means of estimating clinical status of oral health among the general population [49]. The self-rated oral health status tool was adopted from previous studies [50, 51]. The Cronbach alpha for the scale for this study population was 0.60.

*Self-rated quality of oral health.* This was a measure of the perception individual had about the level of disability resulting from the oral health problems. Prior studies in Nigeria [25], and other countries [52], have assessed self-rating of individual quality of oral health, and had shown that self-rating of the quality of oral health was a valid, comprehensive indicator of oral health status [53]. Respondents were asked to rate the quality of their oral health on a scale of 1–5 (Very good, good, fair, poor, very poor). Ratings were then categorized into two: good quality of oral health (very good and good) and poor quality of oral health (fair, poor, very poor).

**Independent variables.** *Depressive symptoms.* Depressive symptoms were assessed with the nine-item Patient Health Questionnaire (PHQ-9). The instrument evaluates the severity of depressive symptoms, with responses rated from "0" (not at all) to "3" (nearly every day). Scores are derived by summing up response scores; possible total scores range from 0 to 27, with increasing scores indicative of worsening depressive symptoms [54]. The Kappa reliability score after a one-month test-retest of the questionnaire among young adults in Nigeria was 0.89 [55]. For this study, the Cronbach's alpha was 0.91.

*Generalised anxiety symptoms.* The seven-item Generalised Anxiety Disorder (GAD-7) questionnaire was used to measure anxiety symptoms [56]. The GAD-7 requires participants to rate how often they have been bothered by each of seven core anxiety symptoms, over the preceding 2 weeks. Response categories include "not at all", "several days", "more than half the days", and "nearly every day", scored as 0, 1, 2, and 3 respectively, with the total score ranging from 0 to 21. The total score was derived by summing the response scores. For this study, the continuous score was used in analyses with increasing scores indicative of worsening anxiety, and the Cronbach's alpha was 0.90.

**Covariates.** *Sociodemographic variables.* These were age at last birthday (in years), sex at birth (male, female), level of education completed (no formal education, primary, secondary, university/tertiary), and employment status (employed, student, retired and unemployed).

**Moderators.** *Resilience.* Resilience was assessed using the 10-item Connor-Davidson resilience scale [57]. The items are rated on a five-point Likert scale ranging from 0 ("Not true at all") to 4 ("True nearly all the time"). Possible total scores thus range from 0 to 40, with higher scores representing higher resilience. The Cronbach's alpha for the scale was 0.81 for a population of nurses in Nigeria [58]. For this study, the Cronbach's alpha was 0.87.

*Self-esteem.* Self-esteem was assessed using the 10-item Rosenberg self-esteem scale [59], which measures global self-worth by assessing both positive and negative feelings about oneself. The scale items are answered using a 4-point Likert scale format ranging from strongly agree (4) to strongly disagree (1). Five of the items are positively worded while five are negatively worded, and these were reverse-scored. The total scores are derived from the sum of all 10 items, and total scores ranged from 10 to 40 on a continuous scale with higher scores representing higher self-esteem. The instrument had been validated for use in diverse populations in Nigeria [54, 58] demonstrating satisfactory reliability and construct validity. For this study, the Cronbach's alpha was 0.75.

**Data analysis.** Statistical analyses were conducted with SPSS version 23.0, and moderation analyses was conducted with Model 1 of the PROCESS procedure for SPSS version 4.0 [60]. The data were described using means and standard deviations or as frequencies and percentages. Bivariate analyses were done to explore the associations between oral health problems, quality of oral health, depression and anxiety. Two separate multivariable binary logistic

regression models were specified to determine the associations between the two dependent variables (oral health status and quality of oral health) and the independent variables (depressive symptoms and anxiety symptoms). All models were adjusted for age, sex, level of education, and employment status. Regression coefficients and their 95% confidence intervals (CI) were calculated.

The hypothesised moderation models were tested in eight models using the 5000-bootstrapping approach to investigate the moderating direct and indirect effects of resilience and self-esteem on the associations between depressive and anxiety symptoms on one hand, and oral health problems and quality of oral health on the other. The "PROCESS" macro, model 59, v 4.0 [60] in SPSS version 23 with bias-corrected 95% confidence intervals was used to test the significance of the mediated effects moderated by resilience and self-esteem. Standardized regression coefficients (β) were calculated. Significance was set at the 5% level, and significant effects were supported by the absence of zero within the confidence intervals.

## Results

Data from all 2,757 respondents who had complete responses were extracted for this analysis. The mean (standard deviation—SD) age of respondents was 20.91 (5.56) years, with age range between 13 and 62 years. Table 1 shows that 1,715 (62.2%%) respondents were male, 1,268 (45.7%) had tertiary level of education, 1,583 (57.4%) were students, and 739 (26.8%) were employed. In addition, there were 1419 (51.5%) 13-19-year-olds. Also, 2,062 (74.8%) participants reported oral health problems, and 925 (33.6%) reported poor quality of oral health. The mean (standard deviation—SD) resilience, self-esteem, depressive symptoms, and anxiety symptoms scores were 21.1 (8.30), 15.0 (1.9), 7.58 (5.9) and 6.4 (4.9) respectively. The mean (standard deviation—SD) of the depressive symptoms and anxiety symptoms scores (p<0.001) were significantly higher in patients with self-reported oral health problems (p<0.001 respectively) and self-reported poor quality of oral health (p<0.001 respectively).

Table 2 shows that the higher the level of resilience (AOR: 0.95; 95% CI: 0.94–0.97; p<0.001) and self-esteem (AOR: 0.92; 95% CI:0.88–0.97; p = 0.001), the significantly lower the odds of reporting oral health problems. In addition, the higher the levels of depressive symptoms the significantly higher the odds of depressive symptoms (AOR: 1.05; 95% CI: 1.02–1.08; p<0.001). Furthermore, the higher the levels of depressive symptoms (AOR: 1.04; 95% CI: 1.02–1.07; p<0.001), anxiety symptoms (AOR: 1.08; 95% CI: 1.04–1.11; p<0.001) and self-esteem (AOR:1.06; 95% CI: 1.01–1.11; p = 0.020) the significantly higher the odds of reporting poor quality of oral health.

Table 3 shows that among the eight moderation models, only resilience emerged as a significant moderating variable of the relationship between anxiety and oral health problems (AOR = -0.004; 95% CI: -0.006 –-0.001; p = 0.002), with the association between anxiety symptoms and oral health problems being significantly lower at higher levels of resilience.

## Discussion

Our study findings indicate that depressive and anxiety symptoms may be significantly associated with a higher likelihood of reporting poor quality of oral health, while depressive symptoms may be associated with a higher likelihood of reporting oral health problems. Though both resilience and self-esteem were significantly positively associated with lower odds of reporting oral health problems, only self-esteem was significantly positively associated with higher odds of reporting poor quality of oral health. Also, resilience significantly moderated the association between anxiety symptoms and oral health problems.

**Table 1. Associations between sociodemographic variables, anxiety, depression, oral health problems and poor quality of oral health in Nigeria (N = 2,757).**

| Variables | Total N (%) | Self -reported oral health problems N = 2,757 | | χ²/t-test p-value* | Self-reported poor quality of oral health N = 2,757 | | χ²/t-test p-value |
|---|---|---|---|---|---|---|---|
| | | Yes | No | | Yes | No | |
| | | N = 2,075 | N = 697 | | N = 925 | N = 1,832 | |
| | | n (%) | n (%) | | n (%) | n (%) | |
| **Age in years [mean (SD)]** | 20.91 (5.56) | 20.63 (5.43) | 21.74 (5.85) | 13.65 <0.001 | 21.33 (6.24) | 20.70 (5.17) | 21.70 <0.001 |
| 13-19-years-old | 1419 (51.5) | 1097 (53.2) | 322 (46.3) | 23.55 <0.001 | 445 (48.1) | 974 (53.2) | 13.27 0.004 |
| 20-24-years-old | 779 (28.3) | 590 (28.6) | 189 (27.2) | | 263 (28.4) | 516 (28.2) | |
| 25-49-years-old | 525 (19.0) | 350 (17.0) | 175 (25.2) | | 199 (21.5) | 326 (17.8) | |
| 50-64-years-old | 34 (1.2) | 25 (1.2) | 9 (1.3) | | 18 (1.9) | 16 (0.9) | |
| **Sex** | | | | | | | |
| Male | 1,715 (62.2) | 1,341 (65.0) | 374 (53.8) | 27.84 | 620 (67.0) | 1,095 (59.8) | 13.77 |
| Female | 1,042 (37.8) | 721 (35.0) | 32 46.2) | <0.001 | 305 (33.0) | 737 40.2) | <0.001 |
| **Educational status** | | | | | | | |
| No formal education | 129 (4.7) | 116 (5.6) | 13 (1.9) | 83.42 | 29 (3.1) | 100 (5.5) | 89.59 |
| Primary education | 85 (3.1) | 68 (3.3) | 16 (2.3) | <0.001 | 36 (3.9) | 48 (2.6) | <0.001 |
| Secondary education | 1,290 (46.5) | 1,036 (50.2) | 248 (35.7) | | 540 (58.4) | 744 (40.6) | |
| Tertiary education | 1,268 (45.7) | 842 (40.8) | 420 (60.1) | | 320 (34.6) | 940 (51.3) | |
| **Employment status** | | | | | | | |
| Employed | 739 (26.8) | 524 (25.4) | 215 (30.9) | 18.32 | 279 (30.2) | 460 (25.1) | 21.37 |
| Student | 1,583 (57.4) | 1,181 (57.3) | 402 (57.8) | <0.001 | 475 (51.4) | 1,108 (60.5) | <0.001 |
| Retired | 21 (0.8) | 18 (0.9) | 3 (0.4) | | 9 (1.0) | 12 (0.7) | |
| Unemployed | 414 (15.0) | 339 (16.4) | 75 (10.8) | | 162 (17.5) | 252 (13.8) | |
| **Depressive symptoms [mean (SD)]** | 7.58 (5.92) | 8.21 (5.61) | 5.60 (5.75) | 3.35 <0.001 | 9.89 (5.27) | 5.63 (5.60) | 38.07 <0.001 |
| **Anxiety symptoms [mean (SD)]** | 6.42 (4.87) | 6.87 (4.64) | 5.00 (4.82) | 4.07 <0.001 | 8.34 (4.26) | 4.76 (4.51) | 43.93 <0.001 |

*Significance set at p<0.05 SD: standard deviation

One of the strengths of the current study is the large sample size of adolescents and adults. The respondent-administered and online questionnaire reduced the risk for social desirability bias. However, we have some limitations: the use of online recruitment meant that the study inadvertently excluded participants who did not have access to smart phones or internet access. This creates a bias sampling of respondents in the higher socioeconomic strata, thereby reducing generalisability of the study. Another study limitation is the cross-sectional descriptive nature of the study design, which limits the ability to affirm causality between the variables explored. Furthermore, the effect sizes of the bivariate associations between the oral health problems, quality of oral health, anxiety symptoms and depressive symptoms were small, indicating that there likely are other variables that also explain the risk for oral health problems and poor quality of oral health. We also used a single question to measure the self-reported quality of oral health. Though single item measures of subjective health and wellbeing did not necessarily correlate with medical diagnoses, the former however, had greater validity when predicting help seeking behaviour and health service use [61].

Relationships between depressive symptoms and poor quality of oral health and oral health problems have been reported in prior studies [11, 62]. Depressive symptoms tend to indirectly interfere with dental care, leading to worsening oral health status and poor quality of oral

**Table 2. Multivariable binary logistic regression analysis to determine factors associated with oral health problems and poor quality of oral health in Nigeria (N = 2,757).**

| Variables | Self-reported oral health problems AOR (95% CI) N = 2,757 | p-value | Self-reported poor quality of oral health AOR (95% CI) N = 2,757 | p-value |
|---|---|---|---|---|
| **Age in years** | | | | |
| 13-19-years-old | 1.00 | - | 1.00 | - |
| 20-24-years-old | 1.16 (0.91–1.49) | 0.239 | 1.33 (1.07–1.66) | 0.012 |
| 25-49-years-old | 0.87 (0.65–1.18) | 0.371 | 2.03 (1.53–2.69) | <0.001 |
| 50-64-years-old | 1.01 (0.44–2.34) | 0.968 | 3.56 (1.70–7.47) | 0.001 |
| **Sex** | | | | |
| Female | 1.00 | - | 1.00 | – |
| Male | 1.62 (1.35–1.95) | <0.001 | 1.36 (1.14–1.62) | 0.001 |
| **Educational status** | | | | |
| No formal education | 1.00 | - | 1.00 | - |
| Primary education | 0.59 (0.26–1.33) | 0.205 | 4.01 (2.15–7.49) | <0.001 |
| Secondary education | 0.62 (0.34–1.13) | 0.115 | 3.66 (2.34–5.70) | <0.001 |
| Tertiary education | 0.44 (0.24–0.81) | 0.008 | 1.50 (0.95–2.39) | 0.085 |
| **Employment status** | | | | |
| Not employed | 1.00 | - | 1.00 | - |
| Employed | 1.16 (0.91–1.46) | 0.233 | 0.86 (0.70–1.07) | 0.181 |
| **Depressive symptoms** | 1.05 (1.02–1.08) | <0.001 | 1.04 (1.02–1.07) | 0.001 |
| **Anxiety symptoms** | 1.01 (0.97–1.04) | 0.728 | 1.08 (1.04–1.11) | <0.001 |
| **Resilience** | 0.95 (0.94–0.97) | <0.001 | 0.99 (0.98–1.00) | 0.142 |
| **Self-esteem** | 0.92 (0.88–0.97) | 0.001 | 1.06 (1.01–1.11) | 0.020 |
| Cox & Snell $R^2$ | 0.093 | - | 0.118 | - |
| Nagelkerke $R^2$ | 0.137 | - | 0.163 | - |
| Omnibus test of model coefficients | 268.25 | <0.001 | 344.88 | <0.001 |
| Hosmer Lemeshow test | 5.41 | 0.713 | 21.53 | 0.006 |

AOR: adjusted odds ratio; CI: confidence interval; SD: standard deviation

**Table 3. Outcomes of moderation analysis exploring the role of resilience and self-esteem as moderators in the effect of the independent variables (anxiety and depression) on the dependent variables (oral health problem and poor oral health quality) (N = 2,757).**

| Independent-Dependent variables | Moderators | AOR (95% CI) | P-value |
|---|---|---|---|
| **Anxiety- Oral health problems** | | | |
| | Resilience | -0.004 (-0.006 – -0.001) | 0.002 |
| | Self-esteem | 0.01 (-0.004 – 0.20) | 0.059 |
| **Depression–Oral health problems** | | | |
| | Resilience | -0.002 (-0.004 – 0.0002) | 0.082 |
| | Self-esteem | 0.003 (-0.005 – 0.012) | 0.447 |
| **Anxiety-Poor oral health quality** | | | |
| | Resilience | -0.0004 (-0.003 – 0.002) | 0.720 |
| | Self-esteem | -0.007 (-0.016 – 0.003) | 0.154 |
| **Depression-Poor oral health quality** | | | |
| | Resilience | 0.003 (-0.002 – 0.002) | 0.765 |
| | Self-esteem | -0.005 (-0.012 – 0.003) | 0.260 |

AOR: adjusted odds ratio; CI: confidence interval; SD: standard deviation

health. Depression is associated with reduced energy and motivation, which can result in neglect of oral hygiene, avoidance of necessary dental care and increased consumption of cariogenic diets. In addition, there is the risk for antidepressant-induced xerostomia. Depression can also negatively impact nutritional intake resulting in poorer oral health status and outcomes [62]. These collectively increase the risk of dental caries and periodontal diseases leading to tooth loss [63]. Thus, the higher odds of reporting poor oral health and poor quality of oral health among people with higher-level depressive symptoms observed in this study was not unexpected.

Unlike depressive symptoms, anxiety symptoms were only associated with self-reporting of poor oral health quality. However, like depressive symptoms, higher levels of anxiety increased the odds of reporting poor oral health quality. Although depressive and anxiety symptoms have some symptom similarity, the two phenomena appear to affect self-reporting of oral health in different ways. Anxiety seems to increase the risk for reporting only poor quality of oral health without increasing the risk for reporting oral health problems. This may be because individuals with anxiety can have both realistic appraisals and distorted perceptions of health [64]. We postulate that individuals who have anxiety disorders can objectively assess their oral health problems but have an altered morbid perception on the impact these problems have on their oral health quality of life. Further studies are needed to test this assumption.

Furthermore, for respondents who reported anxiety symptoms, resilience significantly moderated the reporting of oral health problems; and this effect is inverse–the higher the resilience, the lower the likelihood of reporting oral health problems by those reporting anxiety. As a coping attribute, resilience may not increase the risk for distorting perception. Rather, it enables people to cope with the negative realities and events of their lives [64]. Resilience may reduce the negative impact of the relationship between depression, anxiety, oral health problems and the quality of oral health. However, this was found to be true only for anxiety and oral health problems in this sample. Although we found no significant associations between anxiety and self-reporting of oral health problems, our findings indicate that resilience is another factor that plays a significant role in the interacting pathways between mental wellness and oral health. Resilience may therefore be an important confounding factor for future studies evaluating associations between mental health and oral health.

What we have demonstrated is a complex relationship between mental health, oral health, resilience and self-esteem. The relationships between the variables and their inter-relational roles are not linear, yet the roles of each variable in that network of inter-relationships are vital and can influence oral health. Future comprehensive studies on these relationships are needed to improve our understanding on the relationships between mental and oral health; and how these relationships can help improve the quality of oral health care provided.

In conclusion, the present study indicated that depressive and anxiety symptoms may affect individuals' reporting of oral health status, with resilience attenuating the negative reporting of oral health problems in individuals with anxiety symptoms. These results suggest that studies that assess self-reports of oral health problems and the quality of oral health should include depression, anxiety, resilience and self-esteem as possible confounders. Furthermore, enhancing resilience may indirectly improve oral health status, which, in turn, may enhance individual quality of life.

## Supporting information

**S1 Data.**
(CSV)

## Acknowledgments

We appreciate all the participants, who provided data and contributed their time to make this study possible. The authors also acknowledge the THEDI organization, which conducted the primary study and provided access to the data for this study.

## Author Contributions

**Conceptualization:** Olanrewaju Ibikunle Ibigbami.

**Data curation:** Olanrewaju Ibikunle Ibigbami.

**Formal analysis:** Olanrewaju Ibikunle Ibigbami.

**Methodology:** Olanrewaju Ibikunle Ibigbami.

**Project administration:** Morenike Oluwatoyin Folayan, Olakunle Oginni.

**Supervision:** Morenike Oluwatoyin Folayan, Olakunle Oginni, Nadia A. Sam-Agudu.

**Validation:** Joanne Lusher, Nadia A. Sam-Agudu.

**Writing – original draft:** Olanrewaju Ibikunle Ibigbami.

**Writing – review & editing:** Olanrewaju Ibikunle Ibigbami, Morenike Oluwatoyin Folayan, Olakunle Oginni, Joanne Lusher, Nadia A. Sam-Agudu.

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
