## [Decision Letter · Decision Letter 0]

18 Oct 2022

PONE-D-22-17490The Moderating Effects of Resilience and Self-esteem on Associations between Self-Reported Oral Health Problems, Quality of Oral Health, and Mental Health among Residents in NigeriaPLOS ONE

Dear Dr. Ibigbami,

Thank you for submitting your manuscript to PLOS ONE. After careful consideration, we feel that it has merit but does not fully meet PLOS ONE’s publication criteria as it currently stands. Therefore, we invite you to submit a revised version of the manuscript that addresses the points raised during the review process.

The reviwers have especial concerns on the theorectical model that guied your analyses, as well as the rationale for your study. So, in your revision version, I encourage you to provide an indeep explanation on hypothetical mechanism through which resilience and/or self-esteem moderates the association between depression and anxiety.

We look forward to receiving your revised manuscript.

Kind regards,

Thiago Machado Ardenghi

Academic Editor

PLOS ONE

Journal Requirements:

Reviewers' comments:

Reviewer's Responses to Questions

**Comments to the Author**

1. Is the manuscript technically sound, and do the data support the conclusions?

Reviewer #1: Yes

Reviewer #2: Partly

2. Has the statistical analysis been performed appropriately and rigorously? 

Reviewer #1: No

Reviewer #2: Yes

3. Have the authors made all data underlying the findings in their manuscript fully available?

Reviewer #1: No

Reviewer #2: Yes

4. Is the manuscript presented in an intelligible fashion and written in standard English?

Reviewer #1: Yes

Reviewer #2: No

5. Review Comments to the Author

Reviewer #1: The present study aimed to determine whether depression and anxiety are associated with two subjective oral health outcomes and whether resilience and self-esteem moderate such relationship in a Nigerian sample of individuals with at least 13 years old. The study is interesting and has merit but, in its current form, does not add substantial information to the existing literature. See below my considerations:

1) My main reservation with this manuscript is the conceptualization of statistical analyses. Firstly, the hypothetical mechanism through which resilience and/or self-esteem moderates the association between depression and anxiety with oral health is not clearly described (a DAG would be very useful). Additionally, how authors operationalized the variables made the interpretation quite difficult – especially because these constructs act in opposite directions (exposures vs moderators). In my view, it would be more plausible if authors treated the moderators also as mediators, but in opposite directions. That is, low levels of resilience and self-esteem may be part of the causal path linking depression and anxiety to oral health problems, although a moderating/interactive effect (as authors reported) may also occur. Therefore, both mechanisms should be investigated as they are not mutually exclusive.

Importantly, these relationships are complex and require more structured background in both Introduction and Discussion.

2) Authors should present the hypotheses of the study and a more elaborated rationale in Introduction. They focused mainly on gaps in the literature but a background on the complex relationships assessed is missing.

3) Authors mentioned that the aim of the survey was to identify the magnitude of mental health challenges faced by minority populations. I presuppose in this respect that information on gender (besides sex) was collected; however, only the sex variable was included in the analyses. Since the analyzed sample could also configurates as a strength of the study, I suggest that authors consider the population characteristics when discussing their findings, its relevance, and implications.

4) The way how authors named the outcome variables is also confusing. When they mentioned “self-reported oral health status”, it seems that they are referring to the traditional self-rated/perceived/reported oral health, measured using the global item “In general, how do you rate your oral health (teeth and gums)?” or “how would you describe the health of your teeth or mouth?”. Nevertheless, this seems to be the second outcome, which was named “self-rated quality of oral health”. In this context, authors should specify in the manuscript the question(s) used to collect data on outcome variables and align such definition with the literature.

5) Besides that, I am not convinced on how authors managed the outcome “self-reported oral health status”. There are distinct conditions that may impact differently one’s perception of oral health. This becomes evident when attention is given to the difference in prevalence of the outcomes – although 74.8% of the participants reported at least one oral health problem (varying from swollen gums to missing teeth), 33.6% reported “poor quality of oral health”. Such discrepancy was not mentioned neither explored in the manuscript.

6) The manuscript presents some mismatching references. For instance, lines 94-96, reference 34; and line 187, reference 45.

7) Is there any threshold available for exposures and moderators? Data in a categorical structure may be more easily interpreted, especially for public health purposes.

8) Table 3: Authors reported the regression coefficients instead of odds ratios. Revise accordingly.

9) Lines 297-298: It would be helpful for the scientific community and policymakers if authors provide the implications of this finding.

10) Line 301: Did authors performed any power analysis? None was reported to assume that “subgroup analyses” were robust.

Reviewer #2: The manuscript is interesting, relevant, with a wide population, but some notes need to be revised. Mainly in relation to the theoretical model, variables chosen and justification of the study.

1. The abstract is not very explanatory, it is not clear who the study population is. It doesn't even show results of moderation. In addition, the title shows the moderating effect of resilience and self-esteem on other variables. However, the main ones pointed out in the abstract are logistical and related to opinion. There seems to be an inconsistency between the title, the objective and the results presented.

2. Throughout the text, terms such as "quality of oral health" would be "oral health related quality of life"? If yes, the term quality of oral health is incorrectly used.

3. The text needs an English grammar and linguistic revision.

4. In introduction, justifying that there are no studies in Nigeria on mental and oral health is not a plausible justification. Authors need to show the importance of studying the topic. Moreover, the target population of the study needs to be in the introduction. Why was this population chosen?

5. The study population was aged 13 years old and older. How old were the eldest? what is the age range of the study?

6. There are questions about self-perception of quality of life, what was the reference that the authors based on? Still, how was the quality of life worked, with continuous values?

7. The authors do not explain how the moderation analysis was done; the analysis needs to be detailed.

8. Was an oral health problem considered if the person answered any of the questions or all of the oral health problems? How was oral health considered in the analyses?

9. The theoretical directions between the variables are not very clear. Nor the reason for including so many variables in the study. The authors need to make the choice of variables and their relationships more explicit, as well as the reason for the moderating effects.

10. Still, the authors need to clarify what was considered mental health, it appears loosely in the introduction and discussion, but it is not justified which variables and relationships are part of mental health.

11. There are already more current references for global burden of oral disease, please review them. Likewise, review very old references cited in the text and their formatting, each one is formatted in a different way.

6. PLOS authors have the option to publish the peer review history of their article (what does this mean?). If published, this will include your full peer review and any attached files.

Reviewer #1: **Yes: **Leandro Machado Oliveira

Reviewer #2: No

---

## [Author Response · Author response to Decision Letter 0]

23 Feb 2023

Response to Reviewers

PLOS One Manuscript Number: PONE-D-22-17490

Moderating Effects of Resilience and Self-Esteem on Associations between Self-Reported Oral Health Problems, Oral Health Status, and Mental Health among Adolescents and Adults in Nigeria

17th February 2023

Reviewer #1:

1) My main reservation with this manuscript is the conceptualization of statistical analyses. Firstly, the hypothetical mechanism through which resilience and/or self-esteem moderates the association between depression and anxiety with oral health is not clearly described (a DAG would be very useful). Additionally, how authors operationalized the variables made the interpretation quite difficult – especially because these constructs act in opposite directions (exposures vs moderators). In my view, it would be more plausible if authors treated the moderators also as mediators, but in opposite directions. That is, low levels of resilience and self-esteem may be part of the causal path linking depression and anxiety to oral health problems, although a moderating/interactive effect (as authors reported) may also occur. Therefore, both mechanisms should be investigated as they are not mutually exclusive. Importantly, these relationships are complex and require more structured background in both Introduction and Discussion. 

Response: Thank you for the comments. We understand that there might be more than one approach in the interpretation of the associations among the variables of interest in this context. Hence, we have added a section on the hypothesis guiding our analysis. Please see lines 105-112

2) Authors should present the hypotheses of the study and a more elaborated rationale in the Introduction. They focused mainly on gaps in the literature but a background on the complex relationships assessed is missing.

Response: We have revised the manuscript to address the reviewer’s feedback. Please see lines 109-112

3) Authors mentioned that the aim of the survey was to identify the magnitude of mental health challenges faced by minority populations. I presuppose in this respect that information on gender (besides sex) was collected; however, only the sex variable was included in the analyses. Since the analyzed sample could also configurates as a strength of the study, I suggest that authors consider the population characteristics when discussing their findings, its relevance, and implications.

Response: Thanks for this observation. First, we have edited the described aim of the survey and eliminated the references to sexual minority as this was not the focus of the study. Our focus was to determine the associations between the dependent and independent variables and not the magnitude of the impact. To avoid the table 2 fallacy, we have focused on reporting the associations between our dependent and independent variables and deleted reporting of our confounders for the study. We have however, include age profiling of the study participants in the results as this is an important demographic profile that is important for the study. 

4) The way how authors named the outcome variables is also confusing. When they mentioned “self-reported oral health status”, it seems that they are referring to the traditional self-rated/perceived/reported oral health, measured using the global item “In general, how do you rate your oral health (teeth and gums)?” or “how would you describe the health of your teeth or mouth?”. Nevertheless, this seems to be the second outcome, which was named “self-rated quality of oral health”. In this context, authors should specify in the manuscript the question(s) used to collect data on outcome variables and align such definition with the literature.

Response: We understand the potential for confusion and have now provided some explanations. The approach for the assessment of “self-reported oral health problem” in this study is in line with the approach used by previous studies cited in reference 46 &47. In this context and in that of the cited studies, “self-reported oral health problem” refers to the perception individuals had about having oral health diseases. This is also distinct from “self-rated quality of oral health”, which refers to the perception individual had about the level of disability resulting from the oral health problems. It is also important to emphasize that these were subjective measures which are also valid when predicting help seeking behaviour and health service use [ref]. The explanatory descriptions of self-reported oral health problems vs self-rated quality of health are provided in lines 178-182 and lines 190-194 in the Methods. 

5) Besides that, I am not convinced on how authors managed the outcome of “self-reported oral health status”. There are distinct conditions that may impact differently one’s perception of oral health. This becomes evident when attention is given to the difference in prevalence of the outcomes – although 74.8% of the participants reported at least one oral health problem (varying from swollen gums to missing teeth), 33.6% reported “poor quality of oral health”. Such discrepancy was not mentioned neither explored in the manuscript.

Response: Self-reported quality of oral health may not be a true reflection of actual oral health status. This is also what has been reported in previous studies, and may be related to the fact that other factors play roles in the final outcome. However, perception about one’s self and one’s body image is what powerfully affects mental health status. In addition, the number of oral health problems is not symmetrically aligned with quality of life, and so the discrepancy referred to may be of limited clinical significance. 

6) The manuscript presents some mismatching references. For instance, lines 94-96, reference 34; and line 187, reference 45. Response: We thank the reviewer for noting these errors. The references are now aligned.

7) Is there any threshold available for exposures and moderators? Data in a categorical structure may be more easily interpreted, especially for public health purposes.

Response: Thanks for the question. For logistic regression, continuous data reduce the risk of loss of power. We have chosen to use the measure as a continuous rather than categorical variable as the most appropriate methodology for this study.

8) Table 3: Authors reported the regression coefficients instead of odds ratios. Revise accordingly. Response: Thanks for the observation. This has been corrected.

9) Lines 297-298: It would be helpful for the scientific community and policymakers if authors provide the implications of this finding. Response: The implications of the findings have been discussed; please find them in line 339-345.

10) Line 301: Did authors performed any power analysis? None was reported to assume that “subgroup analyses” were robust. Response: Thanks for raising this query. The reference to robust analysis has been deleted.

Reviewer #2: 

The manuscript is interesting, relevant, with a wide population, but some notes need to be revised. Mainly in relation to the theoretical model, variables chosen and justification of the study.

1) The abstract is not very explanatory, it is not clear who the study population is. It doesn't even show results of moderation. In addition, the title shows the moderating effect of resilience and self-esteem on other variables. However, the main ones pointed out in the abstract are logistical and related to opinion. There seems to be an inconsistency between the title, the objective and the results presented.

Response: Thank you for the very helpful critique. While it makes it a bit longer, we have edited the title to reflect the study objective. We have also made revisions to the Abstract to provide more information and clarity, albeit summarized. Besides indicating the study population as adolescents and adults in Nigeria in the background of the abstract, we have clarified in the Methods section that they were an online population of adolescents and adults, resident in Nigeria aged 13 years and older. The Results indicate that this population ranges from 13 to 62 years of age. The result of the moderation analysis was reported in the results section of the abstract, stating that: Resilience significantly moderated the association between anxiety and oral health problems (AOR= -0.004; 95% CI: -0.006 – -0.001; p=0.002). 

2) Throughout the text, terms such as "quality of oral health" would be "oral health related quality of life"? If yes, the term quality of oral health is incorrectly used. Response: Please see the response to Reviewer 1 Comment 4. We acknowledge the potential for confusion. However, we have provided explanations in the manuscript to address this. We have retained the use of the term ‘quality of oral health’ as this was a subjective measure, as opposed to ‘oral health-related quality of life’, which implies that a more objective measure was done. We have not used these terms interchangeably.

3) The text needs an English grammar and linguistic revision. Response: We appreciate the feedback; we have worked extensively on both grammar and language in the revised manuscript. We have highlighted these edits in red. 

4) In introduction, justifying that there are no studies in Nigeria on mental and oral health is not a plausible justification. Authors need to show the importance of studying the topic. Moreover, the target population of the study needs to be in the introduction. Why was this population chosen?

Response: The target study populations are adolescents and adults in Nigeria. We have updated the justification for selecting this target population: The prevalence of psychological distress among adults in Nigeria is as high as 69.9% [34]. Also, between 28.5% and 68.4% of adults have untreated oral health conditions [35]. Nonetheless, there is a lack of evidence on the mechanisms by which mental health may affect individual perceptions of oral health problems, the negative impact of these problems on one’s quality of life and if these interactions can increase the uptake of dental care services for the treatment of oral health conditions. This is despite the evidence of the strong predictive power of self-reported health for prospective health and social outcomes [36].

5) The study population was aged 13 years old and older. How old were the eldest? what is the age range of the study? Response: Age ranged between 13 and 62 years. This has been added to the Results section, along with the mean age and proportions represented by adolescents and young adults. 

6) There are questions about self-perception of quality of life, what was the reference that the authors based on? Still, how was the quality of life worked, with continuous values

Response: The self-rated oral health status tool is a single question that was developed for the study. It was a single question that could not be validated using Cronbach’s alpha. The content validation was however done as described in the methods section of the study. The strength of the single question to measure the quality of the oral health was discussed in the study limitation. See lines 304-308.

7) The authors do not explain how the moderation analysis was done; the analysis needs to be detailed. Response: Thanks for raising this query. We have expounded on the analysis conducted. Please see lines 243-250

8) Was an oral health problem considered if the person answered any of the questions or all of the oral health problems? How was oral health considered in the analyses? Response: Oral health problems was reported as a binary yes (has oral health problems) or no (had no oral health problems). Please see Methods lines 178-188 for the description in question.

9) The theoretical directions between the variables are not very clear. Nor the reason for including so many variables in the study. The authors need to make the choice of variables and their relationships more explicit, as well as the reason for the moderating effects.

Response: We appreciate the reviewer’s feedback. Our narrative in the Introduction section, lines 94 to 104, provides the theoretical framework and justification for why we included the variables that we analyzed in our paper. It reads: “In addition, resilience and self-esteem can protect against adverse mental health outcomes following adverse exposures, and moderates the association between poor oral health and mental health difficulties [37]. Resilience describes an individual’s innate ability to positively adapt in the face of stress and adversity [37]. On the other hand, self-esteem, which is the evaluative and affective dimension of the self-concept (a cognitive schema that sums one’s beliefs and knowledge about their personal attributes and qualities) [38], affects reactions to stressful events and how individuals cope with stress [39]. Thus, resilience can minimize the impact of health-related adversities [40, 41], including those arising from poor oral health [42, 43], thereby attenuating the associations between poor oral health and mental health disorders like depression and anxiety [44]. There is however, little known about how individual resilience and self-esteem moderate oral health in Nigeria”.

We trust that the above explanation provides the reader a satisfactory justification for the variables explored. 

10) Still, the authors need to clarify what was considered mental health, it appears loosely in the introduction and discussion, but it is not justified which variables and relationships are part of mental health.

Response: The mental health issues specifically assessed and discussed are anxiety and depression. Please see lines 105-107 in the Introduction section We have revised the paper to strengthen our discussion on the association between mental health and oral diseases by citing references reflecting the literature that has established links between depression, anxiety and oral health problems. 

11) There are already more current references for global burden of oral disease, please review them. Likewise, review very old references cited in the text and their formatting, each one is formatted in a different way.

Response: Thanks for the observation. We have worked to harmonize the reference formatting and we have updated the references to reflect more current data.

---

## [Decision Letter · Decision Letter 1]

26 Apr 2023

Moderating Effects of Resilience and Self-Esteem on Associations between Self-Reported Oral Health Problems, Quality of Oral Health, and Mental Health among Adolescents and Adults in Nigeria

PONE-D-22-17490R1

Dear Dr. Ibigbami,

We’re pleased to inform you that your manuscript has been judged scientifically suitable for publication and will be formally accepted for publication once it meets all outstanding technical requirements.

Kind regards,

Thiago Machado Ardenghi

Academic Editor

PLOS ONE

Additional Editor Comments (optional):

Reviewers' comments:

Reviewer's Responses to Questions

**Comments to the Author**

1. If the authors have adequately addressed your comments raised in a previous round of review and you feel that this manuscript is now acceptable for publication, you may indicate that here to bypass the “Comments to the Author” section, enter your conflict of interest statement in the “Confidential to Editor” section, and submit your "Accept" recommendation.

Reviewer #2: All comments have been addressed

Reviewer #3: All comments have been addressed

2. Is the manuscript technically sound, and do the data support the conclusions?

Reviewer #2: Yes

Reviewer #3: Yes

3. Has the statistical analysis been performed appropriately and rigorously? 

Reviewer #2: Yes

Reviewer #3: Yes

4. Have the authors made all data underlying the findings in their manuscript fully available?

Reviewer #2: Yes

Reviewer #3: Yes

5. Is the manuscript presented in an intelligible fashion and written in standard English?

Reviewer #2: Yes

Reviewer #3: Yes

6. Review Comments to the Author

Reviewer #2: The authors answered and contemplated all the questions, making the manuscript clearer. Only one question remained doubtful. Throughout the text there appear words like "path" and "pathway". Pathway analysis refers to mediation analysis, different from the analysis proposed in the article, which was moderation analysis. I suggest revising the whole text, modify the terms and concepts of the analyses, referring to the research question of the manuscript. Moreover, in table 1, there is a value of the depression variable with no percentage.

Reviewer #3: (No Response)

7. PLOS authors have the option to publish the peer review history of their article (what does this mean?). If published, this will include your full peer review and any attached files.

Reviewer #2: No

Reviewer #3: **Yes: **Jessica Klöckner Knorst

---

## [Editor Report · Acceptance letter]

5 May 2023

PONE-D-22-17490R1 

Moderating Effects of Resilience and Self-Esteem on Associations between Self-Reported Oral Health Problems, Quality of Oral Health, and Mental Health among Adolescents and Adults in Nigeria 

Dear Dr. Ibigbami:

I'm pleased to inform you that your manuscript has been deemed suitable for publication in PLOS ONE. Congratulations! Your manuscript is now with our production department. 

Kind regards, 

on behalf of

Dr. Thiago Machado Ardenghi 

Academic Editor

PLOS ONE